# Cannabidiol (CBD) Protects Lung Endothelial Cells from Irradiation-Induced Oxidative Stress and Inflammation In Vitro and In Vivo

**DOI:** 10.3390/cancers16213589

**Published:** 2024-10-24

**Authors:** Lisa Bauer, Bayan Alkotub, Markus Ballmann, Morteza Hasanzadeh Kafshgari, Gerhard Rammes, Gabriele Multhoff

**Affiliations:** 1Department of Radiation Oncology, TUM School of Medicine and Health, University Hospital of the Technical University of Munich (TUM), 81675 Munich, Germany; lisa.bauer@tum.de; 2Radiation Immuno-Oncology Group, Central Institute for Translational Cancer Research (TranslaTUM), TUM School of Medicine and Health, University Hospital of the Technical University of Munich (TUM), 81675 Munich, Germany; morteza.kafshgari@tum.de; 3Institute of Biological and Medical Imaging (IBMI), Helmholtz Zentrum München (HMGU), 85764 Neuherberg, Germany; bayan.alkotub@helmholtz-munich.de; 4Chair of Biological Imaging at the Central Institute for Translational Cancer Research (TranslaTUM), School of Medicine and Health, University Hospital of the Technical University of Munich (TUM), 81675 Munich, Germany; 5Department of Anesthesiology and Intensive Care Medicine, TUM School of Medicine and Health, University Hospital of the Technical University of Munich (TUM), 81675 Munich, Germany; markus.ballmann@tum.de (M.B.); g.rammes@tum.de (G.R.)

**Keywords:** cannabidiol (CBD), inflammation, oxidative stress, radiation-induced lung disease, radiotherapy

## Abstract

Radiotherapy remains a central pillar in the therapy of solid tumors, including lung cancer. However, the clinically applied radiation dose to a lung tumor is limited by the radiation-sensitivity of the surrounding normal tissue such as the lungs and heart. Ionizing irradiation causes reactive oxygen species (ROS)-induced chronic vascular inflammation, which can result in fatal irradiation-induced lung diseases such as pneumonitis and fibrosis. In this study, we demonstrate that cannabidiol (CBD), the non-psychogenic component of cannabis, mediates anti-inflammatory and anti-oxidative effects that protect the microvasculature of the lung against radiation-induced damage using in vitro and in vivo murine models. CBD therefore has the potential to improve the clinical outcome of radiotherapy by reducing normal tissue toxicity in the lung.

## 1. Introduction

Radiation therapy remains integral to the treatment of solid tumors in more than 50% of cancer patients [1,2]. Despite advances in more focused radiation techniques, healthy tissues such as the lungs and heart are partially exposed to significant irradiation doses up to 20 Gy when the tumor resides in the thorax. The lung is particularly sensitive to irradiation-induced damage and chronic fatal radiation-induced lung diseases (RILDs) such as radiation pneumonitis and fibrosis, and therefore can limit the irradiation dose that can be delivered to the tumor [3,4]. As a major mediator of inflammation, the microvasculature contributes to the development of lung complications by disrupting the endothelial barrier function [5], increasing the expression of inflammatory markers, including adhesion molecules [6], and thereby mediating immune cell adhesion at the site of radiation-induced injury. Chronic inflammation and vascular dysfunction can lead to accelerated atherosclerosis [7], fibrosis, and scarring [3], all of which may eventually contribute to the destruction of organ function.

Radiotherapy induces DNA double strand breaks (DSBs) either directly or, in an even more pronounced way, indirectly via the generation of reactive oxygen species (ROS) that react with many subcellular compartments, including mitochondria. A radiation-induced disruption of the mitochondrial membrane potential (MMP) further enhances intracellular ROS levels and eventually triggers apoptosis if the damage is beyond the cell’s capacity to repair it. Severely damaged endothelial cells (ECs) lose their barrier function [5] and initiate inflammatory immune responses (as reviewed in [8,9]).

Given the involvement of sustained inflammation of the microvasculature in the development of fatal RILDs, anti-inflammatory agents are being considered as potential mitigators to reduce radiation-induced damage in lung tissues [3]. However, systemically applied anti-inflammatory drugs might also attenuate the anti-tumor efficacy of ionizing irradiation. In contrast to other anti-inflammatory agents, CBD is a promising drug candidate because, in addition to its anti-oxidative and anti-inflammatory capacity [10], it mediates pro-apoptotic, anti-angiogenic, and anti-metastatic activities against multiple tumor entities [11].

This study investigated the potential of CBD to protect the H5V murine endothelial cell line and primary murine lung ECs in vitro and the lung vasculature in mice from irradiation-induced damage such as oxidative stress and inflammation.

## 2. Materials and Methods

### 2.1. Cell Culture

The H5V murine EC cell line (RRID:CVCL_AZ87) was cultured in high glucose Dulbecco’s Modified Eagle Medium (DMEM) (Sigma-Aldrich, St. Louis, MO, USA) containing 10% *v*/*v* heat-inactivated fetal bovine serum (FBS) (Sigma-Aldrich) and antibiotics (10,000 IU/mL penicillin and 10 mg/mL streptomycin, Sigma-Aldrich). ECs were isolated from non-irradiated and partially irradiated lungs of C57BL/6 mice, as described previously [12]. Lung ECs were cultured on 0.5% *w*/*v* gelatine (Merck, Darmstadt, Germany)-coated surfaces in endothelial cell growth medium (PromoCell, Heidelberg, Germany) containing 10% *v*/*v* heat-inactivated FBS (Sigma-Aldrich) and 1% antibiotics (10,000 IU/mL penicillin and 10 mg/mL streptomycin, Sigma-Aldrich). Primary lung ECs were used for up to 3 passages. All cells were maintained in a humidified atmosphere at 37 °C and 5% *v*/*v* CO_2_.

### 2.2. CBD and NAC Treatment and Irradiation In Vitro

CBD used for in vitro experiments was purchased either as a ready-made solution (Supelco, Sigma-Aldrich) or as a powder (Phytolab, Vestenbergsgreuth, Germany). A stock solution of the powder (1.0 mg/mL in methanol) was freshly diluted in cell culture medium to a final concentration of 10 µM (3.15 µL CBD stock in 1 mL medium). N-acetyl cysteine (NAC, Sigma-Aldrich), a well-known ROS scavenger, was dissolved in de-ionized water (10 mM) and diluted to a final concentration of 100 nM in cell culture medium. The concentration of 100 nM NAC is based on published studies on the survival of endothelial cells after treatment with NAC [13]. Treatment of H5V cells and primary lung ECs from untreated C57BL/6 mice with CBD and NAC in vitro was started 24 h before irradiation and continued until the end of the experiment. The irradiation of cells in vitro was performed on a CIX2 irradiation platform (X-Strahl, Camberley, UK) at a distance of 40 cm with 15 mA, 195 mV, at a dose rate of 1 Gy/45 s.

### 2.3. ROS Measurement

Reactive oxygen species (ROS) were detected using the 2′,7′-dichlorofluorescin diacetate (DCFDA) cellular ROS assay kit (Abcam, Cambridge, UK) following the manufacturer’s instructions. Briefly, cells seeded in flat-bottomed 96-well plates (Costar, Washington, DC, USA) at a density of 0.6 × 10^5^ were treated with CBD and/or NAC or were sham-treated with the diluent alone for 24 h. After incubating the cells with the DCFDA probe for 45 min at 37 °C, the DCFDA solution was removed and replaced with washing buffer supplemented with 10% *v*/*v* FBS and CBD and/or NAC at the indicated concentrations. Fluorescence was measured prior to irradiation, 15 min after irradiation, and 75 min after irradiation on a plate reader (Victor x4, Perkin Elmer, Waltham, MA, USA) at the wavelengths Ex/Em = 485/535 nm. ROS in cells that were cultured in T25 flasks for 24 h in the presence of CBD prior to irradiation were measured 24 h after irradiation. After trypsinizing and counting, cells were incubated with the DCFDA probe for 30 min at 37 °C, and the fluorescence signals were measured using a FACSCalibur™ flow cytometer (BD Biosciences, Franklin Lakes, NJ, USA).

### 2.4. Apoptosis Assay

Cells were incubated in the presence or absence of CBD for the entire duration of the experiment, beginning 24 h prior to irradiation. Apoptotic cells were detected 4 days after irradiation using an Annexin V-FITC apoptosis staining kit (ab14085, Abcam). For this, 2.5 × 10^5^ cells were resuspended in 250 µL Binding Buffer II/1X Binding Buffer, 2.5 µL Annexin V-FITC, and 2.5 µL of propidium iodide (PI) viability stain. Cells were incubated for 5 min in the dark at room temperature (RT) before being analyzed using a FACSCalibur™ flow cytometer (BD Biosciences, Franklin Lakes, NJ, USA).

### 2.5. Western Blotting

Cell pellets of H5V cells and tissues from freshly resected whole lungs from partially irradiated and non-irradiated C57BL/6 mice were lysed in radioimmunoprecipitation assay (RIPA) buffer containing protease cocktail tablets (cOmplete tablets, Roche Diagnostic GmbH, Basel, Switzerland) and phosphatase inhibitors (PhosphoSTOP, Roche Diagnostic GmbH). The protein concentration was determined using the Pierce™ BCA Protein Assay Kit (ThermoFisher Scientific, Waltham, MA, USA). Proteins (20 µg) were separated on an SDS-PAGE (8–15% *v*/*v*) gel and transferred onto activated polyvinylidene fluoride membranes (PVDF), which were then blocked in 1× Roti block buffer for 1 h at RT. Primary antibodies (Table 1) were diluted in 1× Roti block buffer and membranes incubated at 4 °C overnight under rotation. After washing, the membranes were stained with the relevant secondary HRP-conjugated antibodies (Dako-Agilent, Santa Clara, CA, USA; BD Biosciences, Franklin Lakes, NJ, USA; Table 1) diluted in 1× Roti block buffer for 60 min at RT. After washing again, the Pierce™ ECL Western Kit (ThermoFisher Scientific) was used to visualize the protein bands. Images were acquired using the ChemiDoc™ Touch Imaging System (Bio-Rad, Hercules, CA, USA). The protein expression ratio for each sample was quantified using ImageJ Software (version 1.54g).

### 2.6. Cell Cycle Analysis

Cell cycle distribution was determined by flow cytometry. Cells were cultured in the presence or absence of 10 µM CBD for 24 h, trypsinized and fixed (2 × 10^5^ cells) in ice-cold 70% methanol overnight at 4 °C. After washing in PBS, cells were resuspended in PBS containing 0.1% glucose (Sigma-Aldrich) and RNAse (20 µg/mL; New England BioLabs GmbH, Frankfurt, Germany). After 15 min, propidium iodide (PI; final concentration 50 µg/mL) was added and cells were analyzed on a FACSCalibur™ flow cytometer (BD Biosciences). The cell cycle distribution (G0/G1, S, G2/M) was determined based on the intensity of PI signal. The percentage of cells in the cell cycle phases was calculated in FlowJo™ software V10.10 (BD Biosciences).

### 2.7. Animals, CBD Treatment and Irradiation

Animal experiments were performed in accordance with the German Animal Welfare Act (TierSchG) and were undertaken in compliance with the institutional guidelines of the University Hospital of the Technical University Munich. All studies were approved by the Regierung von Oberbayern (licence ROB-55.2-2532.Vet_02-21-195 and ROB-55.2-2532.Vet_02-23-53). Mice were housed under sterile conditions (food and water ad libitum, 23 ± 0.5 °C, 12 h light/dark cycle).

Female 10-week-old C57BL/6 N mice (Charles River, Sulzfeld, Germany) were divided into 3 groups: group 1 was sham-irradiated with 0 Gy (control); group 2 received an image-guided irradiation (16 Gy) of the heart (80%) with a partial lung irradiation (20% of the lung volume) using the Small Animal Radiation Research Platform (SARRP, X-Strahl, Walsall Wood, UK) [14]; group 3 received the same partial lung irradiation (16 Gy) combined with daily injections (i.p.) of CBD (Phytolab) at a concentration of 20 mg/kg body weight [15] dissolved in saline (B. Braun Melsungen, Melsungen, Germany), Tween 20 (Sigma-Aldrich), and DMSO (Sigma-Aldrich) at a ratio of 18:1:1 for a four-week period, starting 2 weeks before irradiation and continuing for another 2 weeks after irradiation. Radiation was performed under isoflurane anesthesia (CP Pharma, Burgdorf, Germany). Mice were sacrificed by cervical dislocation 2 weeks or 10 weeks after irradiation. 

### 2.8. Isolation of Lung Endothelial Cells

Lung tissue from C57BL/6 mice in the three treatment groups (group 1, sham; group 2, partial lung irradiation (16 Gy); group 3, partial lung irradiation combined with a CBD treatment) was collected and digested in collagenase A (Roche) (0.002 g/mL in HBSS (Gibco, Sigma-Aldrich) + 10% *v*/*v* FBS) for 45 min under rotation. The pre-digested tissue was further dissociated by aspirating 10 times through a needle (18 G), followed by filtering through a 70 µm mesh and washing twice in HBSS/10% *v*/*v* FBS. The resulting single-cell suspensions were used for flow cytometry analysis, Western blotting, or isolation of CD31^+^ ECs for further in vitro cell culture, as previously described [12]. For the latter, the single-cell suspensions were incubated with a biotinylated CD31 monoclonal antibody (mAb, MEC13.3, BD Biosciences) which had been labelled using a DSB-X biotin protein labeling kit (Invitrogen, Thermo Fisher Scientific), after which they were incubated with FlowComp™ Dynabeads™ (Invitrogen) in order to separate bead-bound CD31^+^ ECs from other cells via magnetic beads coated to the antibody. CD31^+^ cells were then detached from the beads by incubating with FlowComp™ Release buffer (Invitrogen) and separated from the beads via a magnet, after which they were cultured, as described above.

### 2.9. Isolation of Spleen-Derived Leukocytes

The spleens of C57BL/6 mice of the different treatment groups (group 1, sham; group 2, partial lung irradiation (16 Gy); group 3, partial lung irradiation (16 Gy) combined with a CBD treatment) were collected separately and single-cell suspensions were generated by forcing the tissue through a sterile mesh (70 µm) using a syringe plunger (1 mL) in cold HBSS (Gibco, Sigma-Aldrich) + 10% *v*/*v* FBS. After erythrocyte lysis with erythrocyte lysis buffer (ammonium chloride (Sigma-Aldrich), sodium bicarbonate (Sigma-Aldrich), and 0.5 M EDTA (Gibco), pH 7.2) for 5 min on ice, leukocytes were washed in cold PBS and cryopreserved in 90% FBS/10% DMSO (Sigma-Aldrich) for the leukocyte adhesion assay.

### 2.10. Leukocyte Adhesion Under Flow

Freshly isolated lung ECs of C57BL/6 mice after partial lung irradiation (16 Gy) or partial lung irradiation (16 Gy) combined with a CBD treatment were seeded into the channels of µ-slides VI (0.4 mm ibi-Treat IBIDI, Gräfelfing, Germany) pre-coated with 0.5% *w*/*v* gelatine at a density of 2 × 10^5^ cells per channel. Lung ECs were cultured at 37 °C until a confluent monolayer was reached in each channel. Thawed mouse leukocytes (10 × 10^6^ cells per mL) isolated from spleens of the identically treated C57BL/6 mice were stimulated with Interleukin-2 (IL-2; 100 IU/mL) and the Hsp70-peptide TKD (2 µg/mL) in RPMI medium (Gibco) supplemented with 10% *v*/*v* FBS and 1% *v*/*v* antibiotics (10,000 IU/mL penicillin and 10 mg/mL streptomycin) for 4–5 days in vitro. After fluorescence labelling of the pre-stimulated mouse leukocytes with 5(6)-CFDA SE Cell Tracer kit (Invitrogen, 0.02 µM) following the manufacturer’s instructions, each individual µ-slide channel containing adherent lung EC was perfused for 30 min with the leukocytes (1 × 10^6^) at 37 °C in the dark using the IBIDI pump system (IBIDI) at a physiological relevant flow rate of 1 dyn/cm^2^. Channels were then washed 4 times with PBS to remove non-attached leukocytes and fixed in formaldehyde (3.7% *v*/*v*, Carl Roth GmbH, Karlsruhe, Germany), and leukocyte adhesion was imaged using a Leica Thunder imaging system (Leica Microsystems GmbH, Wetzlar, Germany) in brightfield and at wavelengths Ex/Em = 492/517 nm. Adherent leukocytes per channel were quantified by an automatic counting of fluorescent particles using ImageJ software (version 1.54g). Representative µ-slide channels seeded with murine lung ECs are illustrated in Appendix B Figure A1a. A higher magnification of the analyzed area, including fluorescence-labelled attached leukocytes, is shown in Appendix B Figure A1b.

### 2.11. Flow Cytometry

Cells were washed with Dulbecco’s PBS (DPBS) supplemented with 10% *v*/*v* FBS (‘flow cytometry buffer’), after which they were incubated with a panel of fluorescence-labelled mAbs (Table 2) for 30 min at 4 °C in the dark. Cells were washed, then resuspended in flow cytometry buffer containing PI. Data were then acquired on a MACSQuant^®^ flow cytometer (Miltenyi Biotec, Bergisch Gladbach, Germany) and live (PI-negative) lung ECs (CD31^+^/CD45^−^) analyzed using FlowJo™ software V10.10 (BD Biosciences). Both the percentage of positively stained cells and the mean fluorescence intensity (MFI, median signal intensity) of the expressed markers were determined.

### 2.12. Statistical Analysis

All statistical analyses were performed using GraphPad Prism 10 V10.0.2 (GraphPad Software, Boston, MA, USA). Data are presented as mean ± standard deviation (SD) of at least three independent experiments. Differences between groups were assessed using either one-way ANOVA followed by Tukey’s post hoc test for multiple comparisons or by two-way ANOVA followed by Tukey’s multiple comparisons test for experiments involving more than 2 conditions. Statistical significance was defined as a *p*-value of less than 0.05. In all Figures, significance levels are denoted as follows: * *p* < 0.05, ** *p* < 0.01, *** *p* < 0.001 and **** *p* < 0.0001.

## 3. Results

### 3.1. CBD Decreases Irradiation-Induced ROS Levels in H5V Cells and Murine Lung ECs In Vitro

To study the effects of CBD and N-acetyl cysteine (NAC), a well-known ROS scavenger, on irradiation-induced ROS production [8], the murine endothelial cell line H5V (Figure 1A–C) and primary murine lung ECs (luECs in vitro, Figure 1D) were treated in vitro with non-lethal doses of CBD (10 µM), NAC (100 nM) or with both drugs 24 h before irradiation with 0 Gy (sham), 4 Gy, or 6 Gy. In preliminary experiments using different concentrations ranging from 5 µM up to 20 µM of CBD for the treatment of H5V cells in vitro determined 10 µM as the optimal dose, a concentration of 5 µM of CBD induced only moderate effects, whereas a concentration above 15 µM CBD induced a growth inhibition in H5V cells. ROS levels were measured prior to irradiation (t0) or 15 min (t1) and 75 min (t2) after irradiation with 0, 4, or 6 Gy.

Treating H5V cells with CBD (10 µM), NAC (100 nM), or both drugs for 24 h did not alter the ROS levels at t0 (Figure 1A), but cell culture-induced elevated ROS levels at t1 (15 min, 0 Gy) and t2 (75 min, 0 Gy) were significantly reduced by NAC (100 nM) when used alone or in combination with CBD (10 µM). Like NAC, at t2 (75 min, 0 Gy), treatment of H5V cells with CBD (10 µM) alone or in combination with NAC (100 nM) also significantly decreased ROS levels compared to the elevated levels in the controls (Figure 1A).

Radiation of H5V cells with 4 Gy (Figure 1B) and 6 Gy (Figure 1C) significantly increased the ROS levels at t1 (15 min after irradiation) and t2 (75 min after irradiation), and levels were decreased to pre-irradiation levels after treatment with CBD, NAC, or a combination of the two (Figure 1B,C). These findings demonstrate that CBD has a similar capacity to attenuate oxidative stress in the lung EC line H5V as the ROS scavenger NAC.

The ROS reduction by CBD lasted for at least 24 h after irradiation with 4 Gy, as CBD-treated (10 µM, 48 h) H5V cells had significantly lower ROS levels than their irradiated counterparts (Appendix B Figure A2a). The clonogenic cell survival of H5V cells was not impaired by treatment with CBD alone (10 µM, 48 h), but was significantly decreased 24 h after an in vitro irradiation (4 Gy), whereas CBD treatment (10 µM, 48 h) slightly improved the clonogenic cell survival of irradiated H5V cells (Appendix B Figure A2b).

An in vitro treatment of freshly isolated primary lung ECs (luECs in vitro) with CBD (10 µM) followed by irradiation with 4 Gy revealed no differences in the ROS levels at t0 prior to irradiation (Figure 1D), but irradiation-induced increased ROS levels at t1 (15 min after irradiation) and t2 (75 min after irradiation)—reaching statistical significance at t2—dropped to initial levels with a pre-treatment with CBD (10 µM, 24 h; Figure 1D). Due to the limited availability of primary lung ECs, an in vitro irradiation with 6 Gy and treatment with NAC was not possible.

These data indicate that primary lung ECs respond to CBD treatment in vitro in a similar manner to the lung EC cell line H5V with respect to a reduction in ROS levels.

### 3.2. CBD Protects H5V Murine Endothelial Cells from Irradiation-Induced DNA Double Strand Breaks and Apoptosis In Vitro

Elevated ROS levels can cause DNA double strand breaks and consequently induce apoptosis. We determined DNA double strand breaks by quantifying γH2AX protein levels in H5V cells after an in vitro treatment with CBD (24 h, 10 µM), irradiation (4 Gy), and CBD treatment followed by irradiation (4 Gy) using Western blot analysis, and apoptosis by flow cytometry based on Annexin V/PI staining. As shown in Figure 2A, treatment with CBD did not alter γH2AX levels (relative to β-actin) compared to the control, whereas shortly (15 min) after an irradiation with 4 Gy, levels of γH2AX were significantly increased. Pre-treatment of H5V cells with CBD (10 µM) for 24 h before irradiation (4 Gy) decreased γH2AX levels, although this decrease was not of statistical significance. In line with these results, a treatment of CBD (10 µM) for 120 h did not induce any significant necrosis or apoptosis in H5V cells (Figure 2B,C), whereas irradiation (4 Gy) induced a significant increase in early, but not late, apoptotic cells compared to control and CBD-treated cells (Figure 2C). A pre-treatment with CBD (10 µM) for 24 h prior to irradiation (4 Gy) significantly reduced early apoptosis (Figure 2C). The percentage of late apoptotic cells was not significantly altered in either group (Figure 2C). A treatment of H5V cells with CBD (10 µM) for 24 h resulted in a significant increase in cells in the G1 phase and a decrease in the G2/M phase compared to untreated control cells (Figure 2D). Since cells in the G1 arrest prior to DNA replication are more resistant to irradiation than cells in the G2/M phase [16], the G1 arrest at the time point of irradiation, 24 h after CBD treatment, might contribute to the observed protective effects of CBD for H5V cells.

To investigate the effect of CBD on the induction of apoptosis in vivo, C57BL/6 mice received a sham irradiation (0 Gy), a partial irradiation (16 Gy) to 20% of the lung volume, or were treated with a daily injection of CBD (i.p. 20 mg/kg body weight) 2 weeks before irradiation and 2 weeks after irradiation (Figure 2E, treatment schedule). Apoptosis of ECs isolated from the lungs of these animals (luECs in vivo) was detected by flow cytometry 2 weeks after a partial irradiation (16 Gy) of the lung by Annexin V/PI staining of CD31^+^ gated lung ECs. In line with the in vitro results, an in vivo partial lung irradiation (16 Gy) increased the percentage of early apoptotic cells, whereas CBD treatment of C57BL/6 mice 2 weeks before and 2 weeks after irradiation reduced the prevalence of apoptotic lung ECs to control levels (Figure 2E,F). Late apoptosis was not altered in either group. Since the data represent the results of only 2 independent experiments, no statistical analysis could be calculated.

Taken together, these findings suggest that CBD is radioprotective by reducing ROS production and early apoptosis in vitro. A similar trend with respect to early apoptosis was observed when mice were treated 2 weeks before and 2 weeks after irradiation with CBD in vivo although statistical significance was not reached.

Similar to primary human umbilical vein ECs (HUVECs) [17], CBD also induces a transient growth arrest in H5V cells. However, this effect was non-toxic and reversible in H5V cells. H5V cells cultured in the presence or absence of CBD for 48 h showed a similar clonogenic cell survival with comparable numbers of colonies on day 7 after seeding, and the radiation-induced drop in colonies could be partially recovered by a CBD treatment for 48 h (Appendix B Figure A2b).

### 3.3. CBD Increases Heme Oxygenase-1 Levels in H5V Cells In Vitro and in Lung Cells In Vivo

Heme oxygenase 1 (HO-1), a mediator of anti-oxidative and cytoprotective effects, is capable of attenuating oxidative and inflammatory stressors to the vasculature [18], and CBD has been reported to induce HO-1 in HUVEC cells [13]. Therefore, we considered the HO-1 expression as a potential mediator of the radioprotective effects of CBD in lung ECs. For this, we determined HO-1 levels in H5V cells by Western blotting 48 h after treatment with CBD (10 µM) and 24 h after irradiation with 4 Gy. As shown in Figure 3A, a CBD treatment or irradiation did not significantly alter HO-1 expression compared to the control (*p* = 0.0887). However, cells receiving a combined treatment of CBD (10 µM) followed by irradiation (4 Gy) exhibited significantly elevated HO-1 levels (Figure 3A). Despite a similar decrease in ROS, a treatment of H5V cells with NAC (100 nM) did not result in an upregulation of HO-1 (Figure A3). This finding indicates that CBD and NAC induce different effects in endothelial cells. We next investigated whether CBD would also induce the expression of HO-1 in a murine model. For this, C57BL/6 mice were sham-irradiated (0 Gy), received a partial irradiation (16 Gy) to 20% of the lung volume, or were treated with a daily injection of CBD (i.p. 20 mg/kg body weight) 2 weeks before irradiation and 2 weeks after irradiation (Figure 3B, treatment schedule). Lysates of the whole resected lungs (irradiated and non-irradiated areas) of C57BL/6 mice were generated 4 weeks after the different in vivo treatments, and comparative HO-1 Western blots were performed. No significant changes in the HO-1 levels were detected in whole lung lysates of C57BL/6 mice after sham or partial lung irradiation with 16 Gy, whereas HO-1 levels were significantly increased in lung lysates of C57BL/6 mice receiving both CBD and irradiation (Figure 3B). As an upregulated expression of HO-1 has been shown to attenuate lung fibrosis [19] and ischemia [20], and protect the heart from atherosclerosis [21], this finding indicates that an increased HO-1 expression might mediate the protective effects of CBD against a partial lung irradiation in vivo.

### 3.4. CBD Reduces Irradiation-Induced Inflammation In Vivo

Reactive oxygen species (ROS), which can only be measured shortly after irradiation due to their short half-life, are known to induce chronic inflammatory effects in the microvasculature [8]. In this study, the expression of VCAM-1, ICAM-1, ICAM-2, and MCAM, all of which are involved in the adhesion and transmigration of leukocytes into injured tissues [22,23,24], on lung ECs was used as a surrogate marker of irradiation-induced inflammation after a partial irradiation of the lung of C57BL/6 mice in vivo. Mice (n = 3–4) were sacrificed 2 and 10 weeks after sham irradiation (0 Gy), partial irradiation of the lung with 16 Gy, or a partial irradiation of the lung (16 Gy) combined with daily i.p. injections of CBD (20 mg/kg body weight) for 4 weeks (2 weeks before and 2 weeks after irradiation), and single-cell suspensions were prepared form freshly resected lungs. The median fluorescence intensity (MFI) of VCAM-1 (Figure 4A), ICAM-1 (Figure 4B), ICAM-2 (Figure 4C), and MCAM (Figure 4D) expression on CD31^+^/CD45^−^ lung ECs (luECs in vivo), and the percentage of positively stained cells was determined by flow cytometry. The gating strategy is illustrated in Appendix B Figure A3.

Although no significant changes in the density (MFI) of VCAM-1 (Figure 4A) and ICAM-1 (Figure 4B) expression were detected 2 and 10 weeks after any treatment (sham irradiation, partial lung irradiation with 16 Gy, partial lung irradiation (16 Gy) combined with a daily i.p. CBD treatment for 4 weeks), the percentage of ICAM-1^+^ lung ECs from C57BL/6 mice receiving both CBD and irradiation was significantly reduced compared to lung ECs (luECs in vivo) from C57BL/6 mice receiving irradiation alone (Figure 4B). The timepoint of an upregulated expression of VCAM-1 and ICAM-1 is dose-dependent, and a higher irradiation dose leads to an earlier and stronger upregulation of ICAM-1 than VCAM-1 [6]. We have also shown previously that the upregulation of VCAM-1 and ICAM-1 expression after partial lung irradiation is restricted to the irradiated lung lobe and does not impact the non-irradiated lung lobe [25]. Since the single-cell suspension used for the flow cytometric analysis involved both irradiated and non-irradiated lung tissues, the upregulation of VCAM-1 and ICAM-1 expression at the studied timepoints—2 and 10 weeks after irradiation—was only moderate. However, the significant decrease in percentage of ICAM-1^+^ lung ECs in C57BL/6 mice receiving a CBD treatment prior to irradiation supports the hypothesis of an attenuated vascular lung inflammation induced by CBD (Figure 4B).

With respect to the inflammatory marker ICAM-2, the significant irradiation-induced increase in its expression density (MFI) 10 weeks after lung irradiation was reversed to control levels by a continuous CBD treatment of C57BL/6 mice before and after irradiation (Figure 4C). In addition, the percentage of ICAM-2^+^ lung ECs (luECs in vivo) was found to be significantly reduced in CBD-treated mice compared to mice receiving a partial lung irradiation alone (Figure 4C). This finding is consistent with previous data from our group [6] which shows an upregulation of ICAM-2 expression 5 to 10 weeks after a partial lung irradiation. The relative early timepoint at which the expression of the inflammatory marker ICAM-2 [25] is upregulated suggests that ICAM-2 plays a particularly pivotal role in the acute pneumonitis-associated phase of RILD that occurs between 4 and 16 weeks post-irradiation of the thorax in C57BL/6 mice [4]. We therefore hypothesize that a continuous treatment of patients with lung cancer before and after irradiation with CBD would reduce radiation-induced acute lung toxicity.

The expression density of MCAM, another inflammatory marker, on lung ECs (luECs in vivo) was already upregulated 2 weeks after partial lung irradiation (*p* = 0.0689) but was reduced to below control levels by a continuous treatment of C57BL/6 mice with CBD (Figure 4D). Ten weeks after a partial lung irradiation of C57BL/6 mice, the percentage of MCAM^+^ lung ECs (luECs in vivo) as well as the intensity of MCAM expression reached initial levels, and a CBD treatment did not affect the MCAM expression at this time point (Figure 4D). This suggests that MCAM expression may be involved in very early inflammatory events. Since MCAM can also modulate angiogenesis [26], the early irradiation-induced increase in the MCAM expression may also reflect an angiogenic response.

As a functional read-out of the increased expression of inflammatory markers, leukocyte adhesion was measured in vitro under flow using the IBIDI system (Appendix B Figure A5a). Leukocyte adhesion was compared between lung ECs (luECs in vivo) derived from C57BL/6 mice after partial lung irradiation (16 Gy) and C57BL/6 mice receiving CBD for 4 weeks (2 weeks before and 2 weeks after irradiation) combined with a partial lung irradiation with 16 Gy. Consistent with the observed downregulation of adhesion molecules such as ICAM-1 and ICAM-2 by a treatment of the mice with CBD for 4 weeks, 10 weeks after irradiation, the leukocyte adhesion decreased when lung ECs isolated from irradiated and CBD-treated C57BL/6 mice (luECs in vivo) were perfused for 30 min with stimulated splenic leukocytes from C57BL/6 mice that had undergone the corresponding 4-week treatment plan, compared to the adhesion to luECs isolated from irradiated mice receiving no CBD treatment (Appendix B Figure A5b). This supports the hypothesis that the downregulation of inflammatory adhesion molecules by CBD has a functional effect on leukocyte attachment to lung ECs.

### 3.5. CBD Reduces the Irradiation-Induced Angiogenesis In Vivo

To study angiogenesis caused by irradiation-induced tissue damage in the lung, the density of VE-cadherin, Endoglin, and Integrin β-3 expression on CD31^+^/CD45^−^ lung ECs (luECs in vivo) from C57BL/6 mice was determined by multiparameter flow cytometry, 2 and 10 weeks after sham irradiation (0 Gy), partial irradiation of the lung with 16 Gy, or from mice receiving CBD daily (i.p. 20 mg/kg body weight) for 4 weeks, 2 weeks before and 2 weeks after a partial lung irradiation with 16 Gy. VE-cadherin, in addition to maintaining vascular integrity [27], also controls angiogenesis by regulating the activity of VEGFR [28]. For example, VE-cadherin expression is upregulated in rapidly proliferating ECs such as those present in repair blastema [12]. As shown in Figure 5A, the intensity (MFI) of VE-cadherin expression was significantly increased 2 weeks after a partial lung irradiation, whereas the percentage of VE-cadherin^+^ lung ECs (luECs in vivo) dropped to initial levels when the animals received a daily CBD treatment (i.p. 20 mg/kg body weight) 2 weeks before and 2 weeks after the partial lung irradiation (Figure 5A).

In contrast, 10 weeks after partial lung irradiation, the MFI of VE-cadherin expression remained unaltered in all treatment groups, irrespective of the intervention (Figure 5A).

Endoglin is a marker of proliferating ECs and is expressed during the repair of damaged vessels [29], but a continuous upregulation of Endoglin impairs vessel stability and can result in a leaky vasculature [30]. Irradiation can upregulate the expression of Endoglin on vascular cells [6]. In line with these findings, the percentage of Endoglin^+^ lung ECs significantly increased 2 weeks after a partial lung irradiation, irrespective of the treatment with CBD (Figure 5B), and remained upregulated up to 10 weeks (Figure 5B). A continuous daily i.p. treatment of C57BL/6 mice with CBD (20 mg/kg body weight), 2 weeks before and 2 weeks after partial lung irradiation (16 Gy), significantly decreased the density of Endoglin expression on lung ECs (luECs in vivo) 10 weeks after a partial lung irradiation (Figure 5B).

The expression of Integrin β-3, a modulator of migration and survival in proliferating ECs during angiogenesis [31], remained unaffected after a partial lung irradiation (16 Gy) and a combined treatment consisting of irradiation and CBD treatment (Appendix B Figure A6a).

CD34 is a ‘stemness’ marker which is expressed on circulating endothelial precursor cells that can be recruited to sites of injury to facilitate repair [32]. As shown in Appendix B Figure A6b, 2 weeks after partial lung irradiation (16 Gy), the expression density (MFI) and percentage of CD34^+^ cells remained unaffected. However, after 10 weeks, the percentage of CD34^+^ lung ECs was significantly reduced by irradiation, with a CBD treatment resulting in a significant upregulation of the CD34 expression density (MFI) and in the percentage of CD34^+^ cells (Appendix B Figure A5b). Wu et al. have shown that HO-1 increases the number of circulating endothelial progenitor cells, thereby improving revascularization and repair after injury [33]. Our data are in line with this finding, as CBD was also found to increase HO-1 in lung ECs (Figure 3).

In summary, we could demonstrate that a daily CBD treatment of mice 2 weeks before and 2 weeks after irradiation can protect lung ECs from radiation-induced damage by attenuating inflammation and by normalizing angiogenic signals (Figure 6).

## 4. Discussion

We have shown that a non-lethal concentration of CBD (10 µM) attenuates the irradiation-induced ROS production induced by an in vitro irradiation of H5V cells with 4 Gy and 6 Gy, similarly to the known ROS scavenger NAC (100 nM). Based on these findings, the effects of CBD (10 µM) were tested using primary murine lung ECs (luECs in vitro) that had been irradiated in vitro with 4 Gy. In accordance with its effects on the H5V cell line, CBD also reduced irradiation-induced ROS production in lung ECs (luECs in vitro) after in vitro irradiation. Moreover, CBD (10 µM) also reduced irradiation-induced DNA double strand breaks—as determined by measuring yH2AX protein levels—and thereby decreased early apoptosis in H5V cells when given 24 h before irradiation. The decrease in apoptosis was also observed in luECs two weeks after irradiation in vivo when CBD treatment was administered.

Previous studies from our group revealed that a daily treatment of mice with CBD for 4 weeks is well tolerated and does not induce any toxic side effects. In the current study, we demonstrated that the expression of HO-1 was significantly upregulated in H5V cells, as well as in whole lung cell lysates after an in vitro and in vivo treatment with CBD. However, it remained to be determined whether the HO-1 expression is directly induced by CBD or indirectly via increased ROS levels. NAC is a ROS scavenger and an antagonistic effect of NAC on an observed response is typically considered to confirm a ROS-induced mechanism. In healthy cells, treatment of HUVECs [13] and vascular smooth muscle cells [35] with CBD induced HO-1, but NAC hindered the induction of HO-1 by CBD in these studies, leading to the conclusion that the CBD-dependent HO-1 induction is dependent on ROS. Contradictory effects are found in lung cancer cells [36] and leukemia cells [37] in which a CBD treatment results in increased ROS levels. The reason why cancer cells respond to a CBD treatment with an increase in oxidative stress, while in normal cells, ROS levels decrease upon CBD treatment [38,39], is not fully understood. Some studies suggest a therapeutic window which determines whether CBD exerts antioxidative, protective effects or results in increased ROS levels which cause apoptosis [13,40]. A dysregulated metabolic [41] and redox [42] state inherent to many cancer cells may contribute to an altered sensitivity of cells towards a CBD treatment with respect to ROS.

CBD has also been shown to mediate anti-oxidative effects in multiple cell types [43]; for example, levels of hydrogen peroxide-induced ROS in HUVECs [38] and amyloid-beta peptide-induced ROS in hippocampal neurons [39] could be ameliorated by CBD. The latter findings are in line with our in vitro data, which show a reduction in radiation-induced ROS levels in the murine H5V lung endothelial cell line and in primary lung ECs (luECs in vitro) by CBD. In H5V cells, the reduction in ROS levels by CBD was comparable to that of NAC, and there was no antagonistic effect after a combined treatment using both drugs.

The anti-oxidative stress protein HO-1 can be induced by any ROS-inducing factor including irradiation or hydrogen peroxide, and, in an effort to understand the underlying mechanism of the protective effects of CBD, we compared the levels of HO-1 in H5V cells after an in vitro treatment with irradiation (4 Gy) and CBD (10 µM), and also in whole lung cell lysates from C57BL/6 mice that had received an in vivo partial lung irradiation (16 Gy) combined with a CBD treatment (i.p. 20 mg/kg body weight per day, 2 weeks before and 2 weeks after irradiation of the lung).

We could show a strong increase in the ROS production (up to 5-fold) 15 min to 24 h after irradiation (4 Gy) in H5V cells, but only a moderate increase in the HO-1 expression after 24 h. Moreover, a combined treatment of CBD and irradiation caused a significant increase in the HO-1 expression in H5V cells after an in vitro treatment, an effect which is associated with reduced ROS levels, and in lung cells isolated from C57BL/6 mice after an in vivo application. In contrast, a NAC treatment which reduces ROS levels in the endothelial cell line was unable to induce the HO-1 expression. This finding indicates that CBD and NAC induce different effects in endothelial cells.

HO-1 also induces anti-inflammatory effects on the vasculature in vivo [18]. In our study, the marked increase in HO-1 expression in lung cells from C57BL/6 mice receiving both CBD and irradiation correlates with an attenuation of the irradiation-induced inflammation, as demonstrated by a decrease in the expression of the inflammatory markers MCAM and ICAM-2 after 2 and 10 weeks. MCAM is involved in the trafficking of monocytes across the endothelial barrier into the tissues [23], where they differentiate to macrophages. In a hindleg ischemia model, the highest numbers of macrophages were detected between 1 and 2 weeks after an irradiation [44], which is in accordance to the upregulated MCAM expression on lung tissue cells 2 weeks after an in vivo irradiation of the lung. A continuous CBD treatment of C57BL/6 mice 2 weeks before and 2 weeks after a partial lung irradiation in vivo (16 Gy) was able to reduce the expression of MCAM to initial levels. Loinard et al. has previously shown the contributing role of monocytes and the derived macrophages on reperfusion of irradiated tissue [44], thereby linking into the other pro-angiogenic functions described for MCAM [26].

ICAM-2 is involved in the extravasation of neutrophils into the tissue, and a blockade of ICAM-2 reduces neutrophil extravasation [24]. A high absolute number of neutrophils, as well as high neutrophil-to-lymphocyte ratios, in lung cancer patients prior to radiotherapy is associated with an increased risk of developing severe radiation-induced pneumonitis [45,46]. Furthermore, radiation has been shown to induce a neutrophil-driven response that promotes metastatic spread into healthy tissue [47]. Lastly, neutrophils greatly contribute to fibrosis under chronic inflammatory conditions [48]. A normalization of the intensity of ICAM-2 expression on irradiated ECs by CBD therefore has the potential to attenuate these adverse effects of neutrophils following radiotherapy.

Investigating the attenuating effects of CBD on VCAM-1 and ICAM-1 expression, the most commonly investigated adhesion molecules, with respect to irradiation-induced inflammation, may require different experimental circumstances such as analyzing a later time point after irradiation or irradiating a larger lung volume to induce a stronger response.

Although the in vitro results indicate a protective effect of CBD by preventing some of the irradiation-induced oxidative damage experienced by the cells, the in vitro conditions cannot model the interplay of different cell types in inflammation in vivo, nor the availability of circulating progenitor cells to contribute to repair. It is possible that CBD exerts protective effects during irradiation which manifest as a reduced inflammation due to reduced damage, and improves post-damage repair and prevents or reduces complications from chronic inflammation. Soares et al. have previously shown direct effects of an upregulation of HO-1 on inflammation, particularly a reduction in VCAM and P-selectin expression, in bovine and porcine aortic ECs in which HO-1 is overexpressed [49].

While MCAM, VE-cadherin, and Endoglin contribute to angiogenesis, and angiogenesis can be therapeutic in the context of repair after injury, VE-cadherin- and Endoglin-associated angiogenesis is more often implicated in cancer angiogenesis [50,51,52]. Indeed, VE-cadherin [53] and Endoglin [54] antibodies have been investigated for their anti-cancer efficacy. The normalization of MCAM, VE-cadherin, and Endoglin expression by CBD supports the anti-angiogenic and anti-proliferative effect of CBD that has been described in vitro for HUVECs, but also in vivo, such as vascularization models in Matrigel™ sponges [17] and reduction of tumor angiogenesis [55].

CD34^+^ endothelial progenitor cells are extensively described in the context of therapeutic angiogenesis [56]. Transplantation of CD34^+^ cells has been shown to mediate anti-inflammatory effects in in vivo animal models, such as inducing regulatory T cells [32] or reducing the release of anti-inflammatory cytokines [57]. Transplanted CD34^+^ cells have also been shown to improve neo-vascularization and reduce fibrosis in rats [58]. The attenuating effect of CBD on MCAM, VE-cadherin, and Endoglin expression and its capacity to increase CD34^+^ ECs could therefore be a contributory factor in the anti-angiogenic anti-tumor effect described for CBD, while also increasing CD34^+^ endothelial progenitor cell-mediated repair.

It should be noted that although the CBD treatment in vivo was terminated 2 weeks after irradiation, a significant downregulation of ICAM-2 and Endoglin was detected even 10 weeks after irradiation. This suggests that a preventive treatment with CBD most likely does not suppress or delay the onset of irradiation-induced inflammation, but instead prevents the development of late symptoms induced by chronic inflammation.

Future studies are necessary to elucidate the exact mechanisms how CBD protects luECs from irradiation-induced damage in vivo. These studies should address immune signaling, cytokines, and the impact of CBD on anti-tumor immune responses. In addition to the protective effects of CBD on normal cells, the potential anti-tumor activity of CBD should be tested in tumor mouse models. With respect to different radiation schemes in patients, it is also important to determine the optimal time window of a CBD treatment to achieve a long-lasting protective effect against chronic vascular inflammation.

## 5. Conclusions

This study shows the potential of CBD to protect the lung microvasculature from radiation-induced damage by reducing ROS levels and early apoptosis, and by attenuating inflammation by normalizing angiogenic signals in the endothelial cell compartment. CBD therapy may therefore provide a promising approach for improving the clinical outcomes of radiotherapy by reducing normal tissue toxicity (RILD) in the lung.

## Figures and Tables

**Figure 1 cancers-16-03589-f001:**
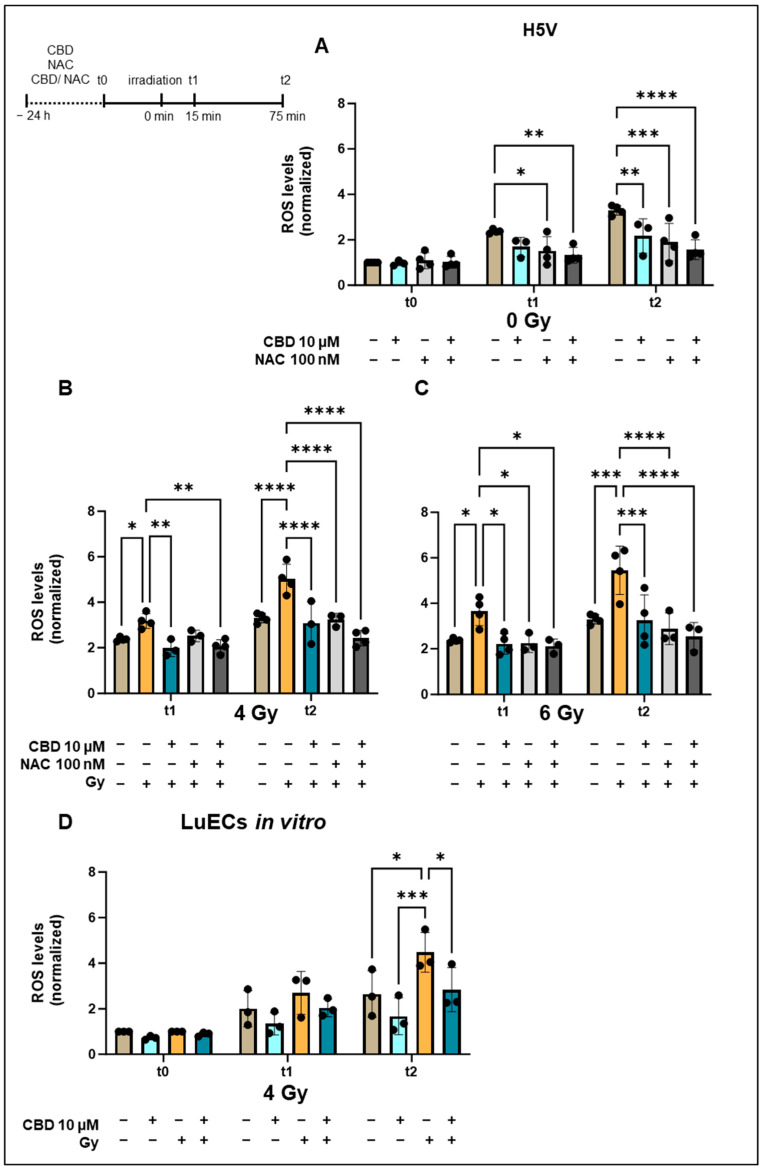
Normalized ROS levels in H5V cells and primary lung ECs (luECs in vitro) upon treatment with CBD, NAC, or both drugs for 24 h followed by an in vitro irradiation (0, 4, 6 Gy). (**A**) ROS levels in H5V cells determined by DCFDA fluorescence measurements after treatment (24 h) with CBD (10 µM), NAC (100 nM), or both drugs at t0, and 15 min (t1) and 75 min (t2) after sham irradiation (0 Gy). (**B**) ROS levels in H5V cells after treatment (24 h) with CBD (10 µM), NAC (100 nM), or both drugs 15 min (t1) and 75 min (t2) after irradiation with 4 Gy. (**C**) ROS levels in H5V cells after treatment (24 h) with CBD (10 µM), NAC (100 nM), or both drugs 15 min (t1) and 75 min (t2) after irradiation with 6 Gy (n = 3–4). (**D**) ROS levels in primary lung ECs (luECs in vitro) after treatment (24 h) with CBD (10 µM) followed by an in vitro irradiation with 0 Gy (sham) and 4 Gy at t0, t1, and t2; (n = 3); all values are normalized to the respective control values at t0; * *p* < 0.05, ** *p* < 0.01, *** *p* < 0.001 and **** *p* < 0.0001; data represent mean ROS levels ± SD.

**Figure 2 cancers-16-03589-f002:**
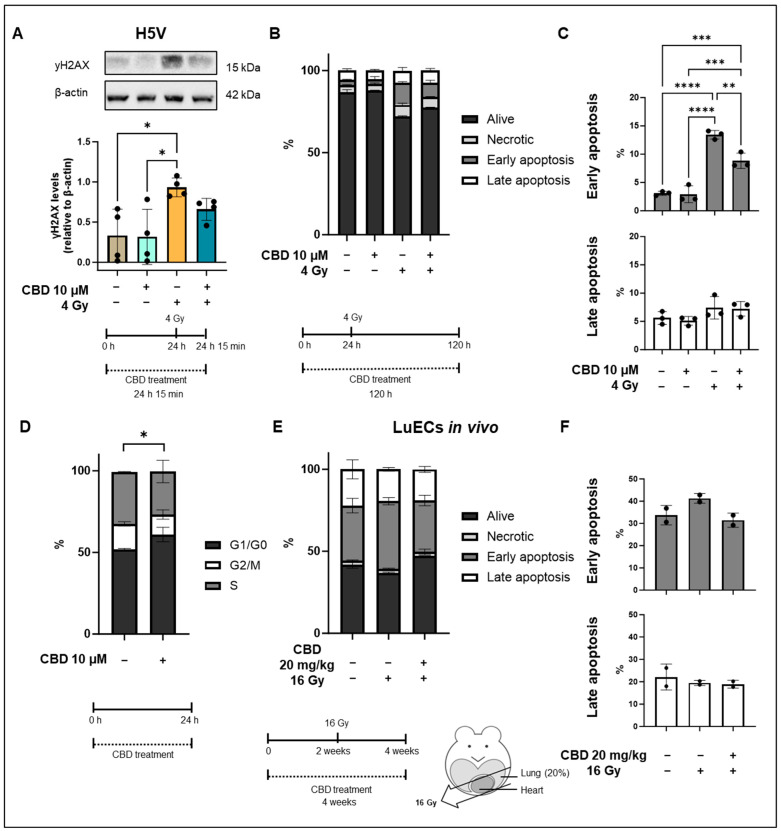
yH2AX levels and apoptosis in H5V cells and primary lung ECs (luECs in vivo) after treatment with CBD and/or irradiation. (**A**) Representative Western blot of yH2AX and relative yH2AX levels in H5V cells that were untreated (first lane), treated with CBD (10 µM, 24 h; lane 2), treated with irradiation (4 Gy; lane 3), or with CBD (10 µM, 24 h) followed by irradiation (4 Gy; lane 4). All yH2AX levels were determined 15 min after irradiation (4 Gy) in H5V cells. The data represent mean values of 4 independent experiments (n = 4). (**B**) Percentage of living (alive), necrotic, early (**C**) and late (**C**) apoptotic H5V cells determined on day 4 by Annexin/PI staining after no treatment (column 1), CBD treatment (10 µM, 24 h; column 2), irradiation (4 Gy; column 3) and a treatment with CBD (10 µm, 24 h) prior to irradiation (4 Gy; column 4). Data are means ± SD from 3 independent experiments (n = 3); * *p* < 0.05, ** *p* < 0.01, *** *p* < 0.001 and **** *p* < 0.0001. (**D**) Cell cycle distribution of H5V cells kept untreated (left column) or 24 h after treatment with 10 µM CBD (right column). Data are means ± SD from 3 independent experiments (n = 3); * *p* < 0.05; significant difference relates to cells in the G1 phase. (**E**) Percentage of living (alive), necrotic, early (**F**) and late (**E**) apoptotic lung ECs (luECs in vivo) determined 2 weeks after in vivo irradiation by Annexin/PI staining after no treatment (column 1), CBD treatment (20 mg/kg body weight per day, 4 weeks; column 2), CBD-treatment (20 mg/kg body weight per day, 4 weeks) prior to irradiation (16 Gy after 2 weeks CBD treatment; column 3). Data are means from 2 independent experiments (n = 2). Original western blots are presented in Appendix A.

**Figure 3 cancers-16-03589-f003:**
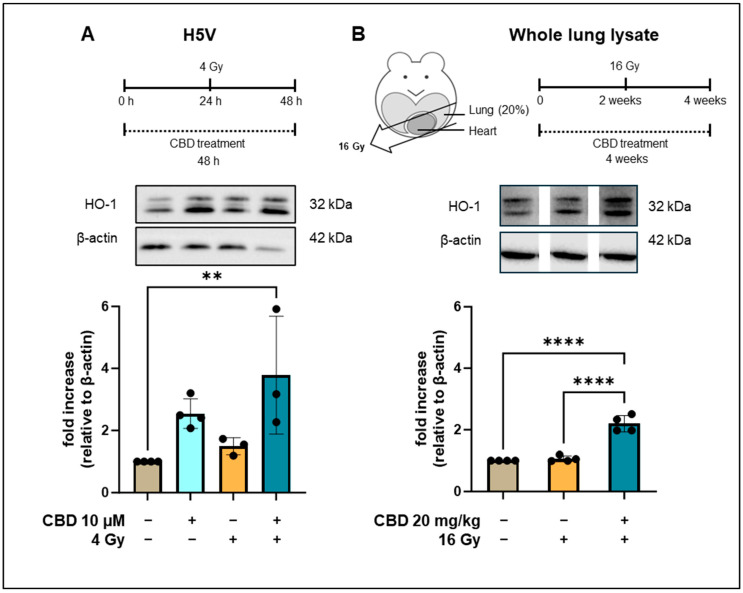
Heme oxygenase-1 (HO-1) levels in H5V and whole lung lysates of C57BL/6 mice. (**A**) Treatment schedule of HV5 cells treated with CBD for 48 h and an irradiation with 4 Gy after 24 h. Representative Western blot of HO-1 and β-actin levels in H5V cells: untreated (lane 1), treated with CBD for 48 h (lane 2), irradiated (4 Gy, 24 h after CBD treatment; lane 3), treated with CBD for 48 h and irradiation (4 Gy, 24 h after CBD treatment; lane 4). The fold increase relative to β-actin is from 3 to 4 independent experiments (n = 3–4). (**B**) Treatment schedule of C57BL/6 mice receiving either CBD for 4 weeks (20 mg/kg body weight per day) or a partial lung irradiation (16 Gy) after 2 weeks. Representative Western blots of HO-1 and β-actin levels (images spliced from same blot) in whole lung lysates of C57BL/6 mice kept untreated (lane 1), after partial lung irradiation (16 Gy, lane 2), or after treatment with CBD (20 mg/kg body weight per day) for 4 weeks and irradiation (partial lung irradiation 16 Gy after 2 weeks, lane 3). Data are mean values ± SD of 4 C57BL/6 mice; ** *p* < 0.01 and **** *p* < 0.0001. Original western blots are presented in Appendix A.

**Figure 4 cancers-16-03589-f004:**
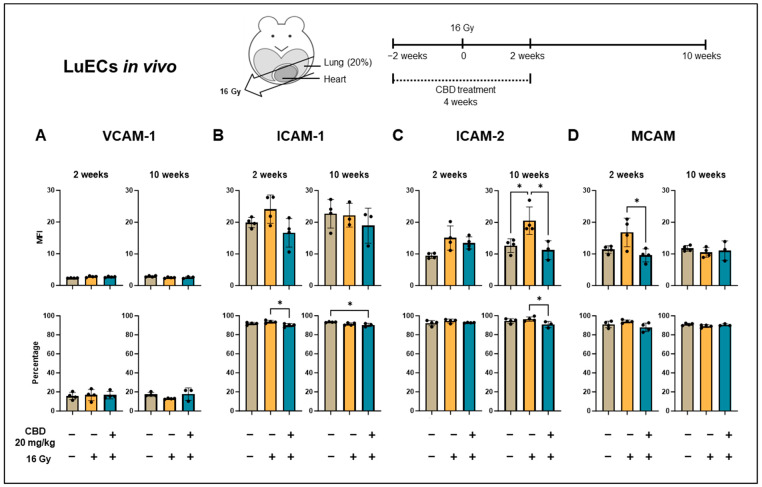
Expression density (mean fluorescence intensity, MFI) and percentage of cells positively stained for inflammatory markers on lung ECs (luECs in vivo) from C57BL/6 mice. (**A**) VCAM-1, (**B**) ICAM-1, (**C**) ICAM-2, and (**D**) and MCAM expression on CD31^+^/CD45^-^ ECs from the lungs of C57BL/6 mice, 2 and 10 weeks after sham irradiation (0 Gy, brown column), partial irradiation of the lung (20% of the total lung volume) with 16 Gy (yellow column), or partial lung irradiation after a treatment with CBD (i.p. 20 mg/kg body weight per day for 4 weeks, 2 weeks before and 2 weeks after lung irradiation, blue column); data show mean values ± SD of 3–4 mice (n = 3–4); * *p* < 0.05.

**Figure 5 cancers-16-03589-f005:**
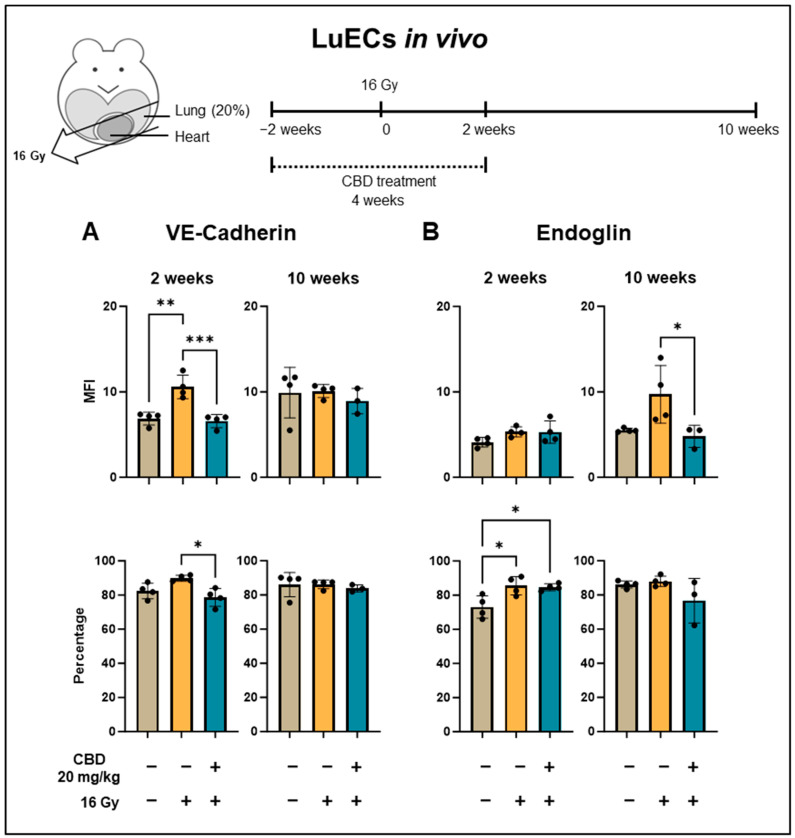
Expression density (mean fluorescence intensity, MFI) and percentage of lung ECs (luECs in vivo) from C57BL/6 mice cells expressing antigens associated with repair and angiogenesis mechanisms. (**A**) VE-cadherin and (**B**) Endoglin expression on CD31^+^/CD45^−^ lung ECs (luECs in vivo) isolated from C57BL/6 mice, 2 and 10 weeks after sham irradiation (0 Gy, brown column), partial irradiation of the lung (20% of the total lung volume) with 16 Gy (yellow column), or partial lung irradiation after a treatment with CBD (i.p. 20 mg/kg body weight per day for 4 weeks, 2 weeks before and 2 weeks after lung irradiation, blue column); data represent mean values ± SD of 3–4 mice (n = 3–4); * *p* < 0.05, ** *p* < 0.01 and *** *p* < 0.001.

**Figure 6 cancers-16-03589-f006:**
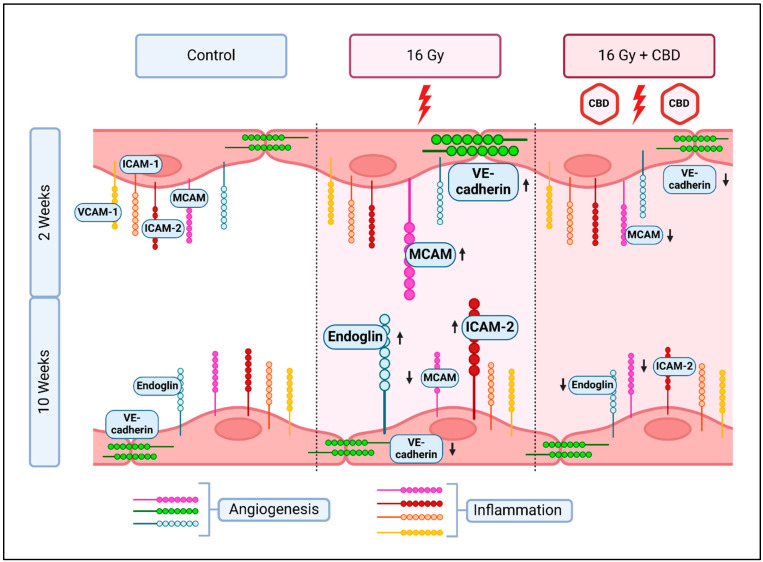
Changes in the expression of inflammatory (VCAM-1, ICAM-1, ICAM-2, MCAM) and angiogenic (VE-cadherin, Endoglin, MCAM) markers on primary lung ECs (luECs in vivo) 2 and 10 weeks after partial irradiation of the lung with 16 Gy, and the recovery effect of a 4-week treatment with CBD, beginning 2 weeks prior to irradiation. Created in BioRender [34].

**Table 1 cancers-16-03589-t001:** Antibodies used for Western blot analysis.

Antibody	Dilution	Company
β-Actin	1:10,000	Sigma-Aldrich
HO-1	1:1000	Cell Signaling Technology, Danvers, MA, USA
yH2AX	1:5000	Abcam
HRP-conjugated rabbit anti-mouse immunoglobulin	1:2000	Dako-Agilent
HRP-conjugated swine anti-rabbit immunoglobulin	1:1000	BD Biosciences

**Table 2 cancers-16-03589-t002:** Fluorescence-labelled monoclonal antibodies.

Antibody	Clone	Dilution	Company
CD31-APC	MEC 13.3	1:4	BD Biosciences
CD34-FITC	RAM34	1:10	eBioscience (ThermoFisher Scientific)
CD45-Vio Blue	REA737	1:10	Miltenyi Biotec
CD54 (ICAM-1)-FITC	3E2	Undiluted	BD Biosciences
CD61 (Integrin β-3)-FITC	HMβ3-1	Undiluted	BD Biosciences
CD102 (ICAM-2)-FITC	3C4 (mIC2/4)	1:10	BD Biosciences
CD105 (Endoglin)-PE	MJ7/18	1:4	eBioscience (ThermoFisher Scientific)
CD106 (VCAM-1)-PE	M/K-2	1:2	Invitrogen (ThermoFisher Scientific)
CD144 (VE-cadherin)-PE	11D4.1	1:4	BD Biosciences
CD146 (MCAM)-PE	ME-9F1	1:4	BD Biosciences

## Data Availability

The original contributions presented in the study are included in the Appendix A; further inquiries can be directed to the corresponding author.

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
