# Peer review of "Cannabidiol (CBD) Protects Lung Endothelial Cells from Irradiation-Induced Oxidative Stress and Inflammation In Vitro and In Vivo"

_cancers, 2024, doi:10.3390/cancers16213589_

Round 1

Reviewer 1 Report

Comments and Suggestions for Authors

This study aims to demonstrate that cannabidiol (CBD) mediates anti-inflammatory and anti-oxidative effects that protect the lung's microvasculature against radiation-induced damage using in vitro and in vivo murine models.

Comments

-Introduction

The introduction is well-written.

The authors stated “anti-inflammatory agents are being considered as potential mitigators to reduce radiation-induced damage in lung tissues [3]. However, systemically applied anti-inflammatory drugs might also attenuate the anti-tumor efficacy of ionizing irradiation”. So, what is the advantage of CBD as it also has an anti-inflammatory capacity?

-Materials and Methods

Materials and methods are detailed and well-written

-Results

-On what basis were the concentrations of CBD (10 µM) and NAC (100 nM)  chosen?

-In Figures 1a, 1b, and 1c, CBD and NAC did not significantly differ in reducing ROS levels following irradiation with 0, 4, and 6 Gy, respectively.

-In Figure 1d, why luECs were not treated with NAC?

-The authors stated “Taken together, these findings suggest that CBD is radioprotective by reducing ROS production, DNA double strand breaks and apoptosis in vitro and in vivo” whereas Pre-treatment with CBD of H5V cells (10 µM) for 24 h before irradiation (4 Gy) did not statistically decrease γH2AX levels and late apoptotic cells (Figure 2a). as well as treatment with CBD it does significantly decrease early and late apoptotic cells in vivo (Figure 2a)????

-In Figure 2, what is the reference for using i.p. injection of CBD at 20 mg/kg body weight?

- Why was NAC not used parallel with CBD (Figure 2) and along the manuscript? Is it expected to get the same results as CBD, or it will be different?

-Discussion

In discussion, the author stated, “a non-lethal concentration of CBD (10 µM) attenuates the irradiation-induced ROS production induced by an in vitro irradiation of H5V cells with 4 Gy 503 and 6 Gy similarly to the known ROS scavenger NAC (100 nM)” (lines 502-503) and  “In cancer cells it is reported that CBD increases ROS levels in tumor cells such as lung cancer cells [32] and leukemia cells [33]” (lines 520-521),   this needs more interpretation.

Author Response

Dear reviewer,

thank you for taking the time to review our manuscript and for your valuable comments. I’d like to address your comments as follows, and have highlighted any changes to the manuscript in yellow:

This study aims to demonstrate that cannabidiol (CBD) mediates anti-inflammatory and anti-oxidative effects that protect the lung's microvasculature against radiation-induced damage using in vitro and in vivo murine models.

 Comments

-Introduction

The introduction is well-written.

The authors stated “anti-inflammatory agents are being considered as potential mitigators to reduce radiation-induced damage in lung tissues [3]. However, systemically applied anti-inflammatory drugs might also attenuate the anti-tumor efficacy of ionizing irradiation”. So, what is the advantage of CBD as it also has an anti-inflammatory capacity?

Response

In contrast to other anti-inflammatory reagents, CBD also has been shown to mediate anti-tumor effects (Hinz, B. and R. Ramer, Cannabinoids as anticancer drugs: current status of preclinical research. Br J Cancer, 2022. 127(1): p. 1-13 DOI: 10.1038/s41416-022-01727-4). This statement has been included into the introduction part. 

-Materials and Methods

Materials and methods are detailed and well-written

-Results

-On what basis were the concentrations of CBD (10 µM) and NAC (100 nM) chosen?

Response

In the literature CBD was used for the treatment of endothelial cells at a concentration of 10 µM. In addition in preliminary experiments in our laboratory we tested different concentrations of CBD ranging from 5 µM up to 20 µM in H5V cells. At a concentration of 5 µM, the CBD mediated effect was only moderate, while concentrations above 15 µM of CBD induced a growth inhibiting effect in H5V cells. Therefore, the optimal concentration of 10 µM of CBD was used in all experiments. This statement has been included in the Results part.

The NAC concentration of 100 nM was chosen based on the effects of NAC on the survival of endothelial cells shown in the literature (Bockmann, S. and B. Hinz, Cannabidiol Promotes Endothelial Cell Survival by Heme Oxygenase-1-Mediated Autophagy. Cells, 2020. 9(7) DOI: 10.3390/cells9071703). This statement has been included in the Materials and Methods part.

-In Figures 1a, 1b, and 1c, CBD and NAC did not significantly differ in reducing ROS levels following irradiation with 0, 4, and 6 Gy, respectively.

Response

At 0 Gy ROS production was only moderately induced by cell culture conditions. This moderate induction of ROS was significantly down-regulated by NAC and a combination of NAC and CBD, and also CBD alone at the later timepoint t2. At 4 and 6 Gy, when ROS was induced by irradiation, CBD and NAC resulted in a comparable reduction in ROS but a combination treatment of CBD and NAC did not cause an additive or synergistic effect with respect to ROS reduction. These data show that no synergistic effects occur when CBD and NAC are given in combination.    

- Why was NAC not used parallel with CBD (Figure 2) and along the manuscript? Is it expected to get the same results as CBD, or it will be different?

Answer

NAC as a ROS scavenger was used as an internal control for the ROS assay shown in Figure 1.

The reason why NAC was not included along the manuscript was predominantly due to the fact that the availability of primary luECs was very limited and because effects of NAC with respect to radioprotection have been described already in the literature:

Mercantepe, T., Topcu, A., Rakici, S. et al. The radioprotective effect of N-acetylcysteine against x-radiation-induced renal injury in rats. Environ Sci Pollut Res 26, 29085–29094 (2019). https://doi.org/10.1007/s11356-019-06110-0

Elham Motallebzadeh, Marwah Suliman Maashi, Mustafa Z. Mahmoud, Akbar Aliasgharzedeh, Zarichehr Vakili, Sayyed Alireza Talaei, Mehran Mohseni, Radioprotective effects of N-acetylcysteine on rats’ brainstem following megavoltage X-irradiations, Applied Radiation and Isotopes, Volume 187,2022, 110348, ISSN 0969-8043, https://doi.org/10.1016/j.apradiso.2022.110348.

Ramune Reliene, Julianne M. Pollard, Zhanna Sobol, Benedicte Trouiller, Richard A. Gatti, Robert H. Schiestl, N-acetyl cysteine protects against ionizing radiation-induced DNA damage but not against cell killing in yeast and mammals, Mutation Research/Fundamental and Molecular Mechanisms of Mutagenesis, Volume 665, Issues 1–2, 2009, Pages 37-43, ISSN 0027-5107, https://doi.org/10.1016/j.mrfmmm.2009.02.016.

Kim HJ, Kang SU, Lee YS, Jang JY, Kang H, Kim CH. Protective Effects of N-Acetylcysteine against Radiation-Induced Oral Mucositis In Vitro and In Vivo. Cancer Res Treat. 2020 Oct;52(4):1019-1030. doi: 10.4143/crt.2020.012. Epub 2020 Jun 18. PMID: 32599978; PMCID: PMC7577823.

Due to the finding that NAC and CBD are not acting additive or synergistically with respect to their inhibitory effects on ROS in H5V cells we hypothesize that NAC might act differently to CBD. This finding is further supported by the finding that CBD but not NAC induces an upregulation of HO-1 in H5V cells either as a single treatment or in combination with irradiation (these data have been included as appendix figure A3).   

-Figure 1d, why luECs were not treated with NAC?

Answer

The reason why primary luECs were not treated with NAC in vitro (Figure 1d) was due to the fact that the amount of primary ECs after isolation from mouse lungs is very limited. Primary luECs can not expanded in vitro because they do not proliferate. For the in vivo experiments we treated mice only with CBD because our Tierversuchsantrag (TVA) only covers this drug treatment. Since it is very time consuming to get permission of the Regierung von Oberbayern to treat mice with an additional drug (more than 12 months) we cannot provide data on an in vivo NAC treatment of mice in this study.

-The authors stated “Taken together, these findings suggest that CBD is radioprotective by reducing ROS production, DNA double strand breaks and apoptosis in vitro and in vivo” whereas Pre-treatment with CBD of H5V cells (10 µM) for 24 h before irradiation (4 Gy) did not statistically decrease γH2AX levels and late apoptotic cells (Figure 2a). as well as treatment with CBD it does significantly decrease early and late apoptotic cells in vivo (Figure 2a)????

Answer

Although the decrease in γH2AX in H5V cells after irradiation and CBD treatment was not significant 15 min after irradiation, in all 4 independent experiments we determined a trend towards lower γH2AX levels compared to the values after irradiation. Based on the comment of the reviewer we revised the statement with respect to the radioprotective effects in vitro and in vivo.

-In Figure 2, what is the reference for using i.p. injection of CBD at 20 mg/kg body weight?

Answer

The concentration of 20 mg/kg body weight of the mouse is based on the literature, showing positive effects on oxidative stress and inflammation in cardiac dysfunction, and generally being within the common range of CBD concentrations used in mouse/rat studies. (Rajesh, M., P. Mukhopadhyay, S. Batkai, V. Patel, K. Saito, S. Matsumoto, Y. Kashiwaya, B. Horvath, B. Mukhopadhyay, L. Becker, et al., Cannabidiol attenuates cardiac dysfunction, oxidative stress, fibrosis, and inflammatory and cell death signaling pathways in diabetic cardiomyopathy. J Am Coll Cardiol, 2010. 56(25): p. 2115-25 DOI: 10.1016/j.jacc.2010.07.033.). This reference is included in the manuscript as number 15.

-Discussion

 In discussion, the author stated, “a non-lethal concentration of CBD (10 µM) attenuates the irradiation-induced ROS production induced by an in vitro irradiation of H5V cells with 4 Gy 503 and 6 Gy similarly to the known ROS scavenger NAC (100 nM)” (lines 502-503) and  “In cancer cells it is reported that CBD increases ROS levels in tumor cells such as lung cancer cells [32] and leukemia cells [33]” (lines 520-521),   this needs more interpretation.

Answer

This aspect has been addressed in the discussion part as follows: “Contradictory effects are found in lung cancer cells [35] and leukemia cells [36] in which a CBD treatment results in increased ROS levels. The reason why cancer cells respond to a CBD treatment with an increase in oxidative stress, while in normal cells ROS levels decrease upon CBD treatment [38, 39], is not fully understood. Some studies suggest a therapeutic window which determines whether CBD exerts antioxidative, protective effects or results in increased ROS levels which cause apoptosis [13, 40]. A dysregulated metabolic and redox state inducing a faster cell growth of cancer cells may contribute to an altered sensitivity of cells towards a CBD treatment.”

(Zhao, Y., E.B. Butler, and M. Tan, Targeting cellular metabolism to improve cancer therapeutics. Cell Death Dis, 2013. 4(3): p. e532 DOI: 10.1038/cddis.2013.60.) (Perillo, B., M. Di Donato, A. Pezone, E. Di Zazzo, P. Giovannelli, G. Galasso, G. Castoria, and A. Migliaccio, ROS in cancer therapy: the bright side of the moon. Exp Mol Med, 2020. 52(2): p. 192-203 DOI: 10.1038/s12276-020-0384-2.)

Reviewer 2 Report

Comments and Suggestions for Authors

The manuscript by Bauer et al. studied the protective effect of cannabidiol on lung endothelial cells in condition of irradiation. The research aimed to find a potential solution to ameliorate the side effects of radiotherapy in cancer patients. With in vitro and in vivo models, the authors showed that cannabidiol reversed the irradiation-altered markers back to control level, including ROS, DNA double strand breaks, apoptosis, inflammation and angiogenesis. Moreover, cannabidiol increased the level of heme oxygenase 1, a cytoprotective mediator.

The introduction provides background on the irradiation-induced lung disease in radiotherapy and the unmet needs – to look for mitigators of the side effects. Experiments were well designed. The results are clearly and thoroughly explained. In discussion, the authors considered the potential of cannabidiol in clinical use to reduce the side effect of irradiation.

One thing I notice is that mechanistic clarification, like molecular pathways, is lacking in the paper. However, the results shown in the manuscript are good and promising enough for publishing and the research area. I would suggest it be accepted.

1, Please correct the following error message in the manuscript. “Error! Reference source not found”.

2, I wonder if the authors have any plan to do any mechanistic exploration in the future. ROS is involved in and interacts with multiple pathways, such as ER stress, autophagy, metabolism, cell cycle, etc. If possible, please briefly discuss potential molecular pathways that the authors consider worthwhile to explore.

Author Response

Dear reviewer,

thank you for taking the time to review our manuscript and for your valuable comments. I’d like to address your comments as follows, and have highlighted any changes to the manuscript in yellow:

Reviewer 2

The manuscript by Bauer et al. studied the protective effect of cannabidiol on lung endothelial cells in condition of irradiation. The research aimed to find a potential solution to ameliorate the side effects of radiotherapy in cancer patients. With in vitro and in vivo models, the authors showed that cannabidiol reversed the irradiation-altered markers back to control level, including ROS, DNA double strand breaks, apoptosis, inflammation and angiogenesis. Moreover, cannabidiol increased the level of heme oxygenase 1, a cytoprotective mediator.

The introduction provides background on the irradiation-induced lung disease in radiotherapy and the unmet needs – to look for mitigators of the side effects. Experiments were well designed. The results are clearly and thoroughly explained. In discussion, the authors considered the potential of cannabidiol in clinical use to reduce the side effect of irradiation.

One thing I notice is that mechanistic clarification, like molecular pathways, is lacking in the paper. However, the results shown in the manuscript are good and promising enough for publishing and the research area. I would suggest it be accepted.

1, Please correct the following error message in the manuscript. “Error! Reference source not found”.

Answer

The erroneous figure cross references have been corrected.

2, I wonder if the authors have any plan to do any mechanistic exploration in the future. ROS is involved in and interacts with multiple pathways, such as ER stress, autophagy, metabolism, cell cycle, etc. If possible, please briefly discuss potential molecular pathways that the authors consider worthwhile to explore.

Answer

In the discussion this aspect of future experiments has been addressed. “Future studies are necessary to elucidate the exact mechanisms how CBD protects luECs from irradiation induced damage in vivo. These studies should address immune signaling, cytokines and the impact of CBD on anti-tumor immune responses. In addition to the protective effects of CBD on normal cells the potential anti-tumor activity of CBD should be tested in tumor mouse models. With respect to different radiation schemes in patients it is also important to determine the optimal time window of a CBD treatment to achieve a long-lasting protective effect against chronic vascular inflammation.”

With respect to the cell cycle, we found that a CBD treatment induces a G1-arrest in H5V cells after 24 h when the cells are irradiated. Since it is well known that G1 cells are better protected from irradiation induced damage this cell cycle arrest could provide an explanation for the protective effect of CBD on endothelial cells. We have included the results of the cell cycle arrest as an additional Figure 2D.

We hope that we could address the helpful comments and suggestions of the reviewers. Thank you for your time and consideration.